# SARS-CoV-2 Accessory Protein Orf7b Induces Lung Injury via c-Myc Mediated Apoptosis and Ferroptosis

**DOI:** 10.3390/ijms25021157

**Published:** 2024-01-18

**Authors:** Rushikesh Deshpande, Wangyang Li, Tiao Li, Kristen V. Fanning, Zachary Clemens, Toru Nyunoya, Lianghui Zhang, Berthony Deslouches, Aaron Barchowsky, Sally Wenzel, John F. McDyer, Chunbin Zou

**Affiliations:** 1Department of Environmental and Occupational Health, Graduate School of Public Health, University of Pittsburgh, Pittsburgh, PA 15213, USA; rhd10@pitt.edu (R.D.); tdesl19@pitt.edu (B.D.); aab20@pitt.edu (A.B.); swenzel@pitt.edu (S.W.); 2Division of Pulmonary, Allergy, Critical Care, and Sleep Medicine, University of Pittsburgh, Pittsburgh, PA 15213, USAkrisvf14@gmail.com (K.V.F.); nyunoyat@upmc.edu (T.N.); zhangl22@upmc.edu (L.Z.); mcdyerjf@upmc.edu (J.F.M.); 3Vascular Medicine Institute, University of Pittsburgh, Pittsburgh, PA 15213, USA

**Keywords:** SARS-CoV-2, COVID-19, apoptosis, ferroptosis

## Abstract

The pandemic of coronavirus disease 2019 (COVID-19) has been the foremost modern global public health challenge. The airway is the primary target in severe acute respiratory distress syndrome coronavirus 2 (SARS-CoV-2) infection, with substantial cell death and lung injury being signature hallmarks of exposure. The viral factors that contribute to cell death and lung injury remain incompletely understood. Thus, this study investigated the role of open reading frame 7b (Orf7b), an accessory protein of the virus, in causing lung injury. In screening viral proteins, we identified Orf7b as one of the major viral factors that mediates lung epithelial cell death. Overexpression of Orf7b leads to apoptosis and ferroptosis in lung epithelial cells, and inhibitors of apoptosis and ferroptosis ablate Orf7b-induced cell death. Orf7b upregulates the transcription regulator, c-Myc, which is integral in the activation of lung cell death pathways. Depletion of c-Myc alleviates both apoptotic and ferroptotic cell deaths and lung injury in mouse models. Our study suggests a major role of Orf7b in the cell death and lung injury attributable to COVID-19 exposure, supporting it as a potential therapeutic target.

## 1. Introduction

Severe acute respiratory syndrome coronavirus 2 (SARS-CoV-2) infection, referred to as coronavirus disease 2019 (COVID-19), resulted in an unprecedented global pandemic, with hundreds of millions of cases and millions of deaths worldwide [1]. SARS-CoV-2 is a novel virus belonging to the *coronaviridae* family, out of which six other members, namely severe acute respiratory syndrome coronavirus (SARS-CoV), middle east respiratory syndrome coronavirus (MERS-CoV), human coronavirus 229E (EHCoV-229E), human coronavirus OC43 (HCoV-OC243), human coronavirus NL63 (HCoV-NL63), and human coronavirus HKU1 (HCoV-HKU1), have been reported to infect humans [2]. The SARS-CoV-2 genome expresses 29 confirmed proteins–four structural proteins (spike (S), envelope (E), membrane (M), and nucleocapsid (N)), 16 non-structural proteins (NSPs 1–16), and nine accessory proteins (ORFs 3a, 3b, 6, 7a, 7b, 8, 9a, 9b, and 10) [3,4]. In general, structural proteins function in cell entry, infectivity, and assembly [3,4]. NSPs are believed to support viral replication, assembly, and secretion [3,4]. The ORF accessory proteins are responsible for evading the host defense and immunity [3,4].

SARS-CoV-2 is a highly transmissible airborne infectious agent [2]. The spike protein of SARS-CoV-2 interacts with angiotensin-converting enzyme 2 (ACE2), which mediates viral entry [3]. The ACE2 protein is expressed in many cell types and many organs, but is highly expressed in ciliated airway epithelia and alveolar type II cells [3]. Although SARS-CoV-2 infection leads to multi-organ damage, acute lung injury and subsequent acute respiratory distress syndrome are the major phenotypes of the disease that appear in the most severe cases [5,6]. SARS-CoV-2 infection causes a strong immune response that leads to hyper-inflammation, exhibiting as a cytokine storm [5,6]. Hyperinflammation defines disease severity, in association with pneumonia and acute respiratory distress syndrome, which increase mortality [5,6]; however, the molecular mechanisms that drive lung injury and acute respiratory distress syndrome have not been fully studied.

Tissue damage is one of the most prominent pathological features of COVID-19, characterized by elevated levels of released levels of lactate dehydrogenase (LDH) or Troponin I [5,6], consistent with cell death. Indeed, post mortems of COVID-19 lungs exhibit massive epithelial and endothelial cell death. SARS-CoV-2 infection has been associated with multiple cell death pathways, including both accidental (necrotic) and regulated cell death [4]. Regulated cell death can be further classified as apoptotic and non-apoptotic [7]. Non-apoptotic regulated cell death type consists of ferroptosis, anoikis, netosis, pyroptosis, and necroptosis (and perhaps others) [7]. Which cell death pathways are the most critical for the development of severe disease and how viruses activate them remains to be elucidated. For example, ferroptosis is a type of cell death involving elevated levels of iron and the strong promotion of lipid peroxidation, which is achieved by the deactivation of GPX4 or its cofactor glutathione (GSH) [8,9]. Production of reactive oxygen species (ROS) on account of lipid autooxidation and the Fenton reaction are also alternative pathways that cause ferroptosis [8,9,10,11].

In this study, we report that the viral accessory protein Orf7b causes remarkable lung epithelial cell death. Orf7b induces both apoptotic and ferroptotic cell death and lung injury that are promoted by c-Myc transactivation.

## 2. Results

### 2.1. Orf7b Promotes Cytotoxicity in Lung Epithelial Cells

To evaluate the cytotoxicity of viral factors, we overexpressed individual viral proteins in lung airway epithelial BEAS-2B cells. After 24 h of transfection, cytotoxicity was evaluated using LDH (Lactate Dehydrogenase) and viable cell counting (Trypan blue staining) assays. Among the viral factors screened, S, M, Orf3a, Orf6, Orf7b, NSP6, and NSP16 all substantially increased LDH release/cell death (Figure 1, upper panel). Similar results were obtained with S, M, Orf3a, Orf6, Orf7b, Orf9b, NSP6, and NSP16 using Trypan Blue staining (Figure 1, lower panel). Orf7b was one of the top-ranked cytotoxic proteins (*p* = 0.02 in the LDH assay and *p* = 0.004 in the Trypan blue assay). These cell death results were confirmed when we transduced Orf7b into human alveolar epithelial carcinoma A549 cells and primary human small-airway epithelial cells (Figure 2A–C and Appendix A). The effect of Orf7b on cytotoxicity was plasmid concentration dependent (Figure 2D) and time-point dependent (Figure 2E). These results indicate that Orf7b is an important viral factor that promotes lung epithelial cell death.

### 2.2. Total RNA seq Analysis Implicates Apoptosis and Ferroptosis, with c-Myc as an Important Transcription Regulator

Multiple cell death pathways might be involved in organ failure due to COVID-19 [12]. Orf7b has been shown to induce apoptosis in Vero-E6 and HEK 293T cells [13,14]. Therefore, we investigated the cell death pathways involved in the cytotoxicity of Orf7b in BEAS-2B cells using total RNA sequencing. IPA (Ingenuity Pathway analysis) predicted the implication of death receptor signaling, ferroptosis, apoptosis, and c-Myc-mediated apoptosis signaling (Figure 3A). Functional enrichment using ENRICHR revealed c-Myc as one of the top transcription regulators (Figure 3B). A graphical summary of IPA showed an indirect correlation between c-Myc and NUPR1 (nuclear protein 1), a critical ferroptosis inhibitor [15,16] (Figure 3C). An upstream network analysis also suggested the association of 12 transcription regulators with c-Myc activation (Figure 3D). In summary, RNA-seq analysis predicted the induction of apoptosis and ferroptosis, and the activation of c-Myc by Orf7b.

### 2.3. Orf7b Induces Both Apoptosis and Ferroptosis

We performed in vitro experiments using lung epithelial cells to test the total RNA seq predictions. Cleaved caspase 8 protein, a marker of extrinsic apoptotic cell death, was upregulated in Orf7b-overexpressing cells (Figure 4A), while no differences were found in the levels of caspase 9, a marker of intrinsic apoptotic cell death, between the control and treatment groups (Figure 4B). This indicated the involvement of the extrinsic apoptotic pathway in Orf7b-induced cell death. Further immunoblotting showed that GPX4, a critical regulator of ferroptosis [8,9], was downregulated in the Orf7b-overexpressing group compared with the control group (Figure 4C). We assessed the malondialdehyde (MDA) levels, and found them to be elevated in Orf7b-transduced cells, suggesting the induction of lipid peroxidation (Figure 4D). We also used 2′,7′-dichlorofluorescin diacetate (DCFDA) for the quantitative assessment of reactive oxygen species (ROS) and found significant ROS levels in the Orf7b-overexpressed group compared with the control group (Figure 4E). These results indicate that Orf7b-induced cytotoxicity also involved ferroptosis. To confirm these observations, we determined the effects of the pan-apoptosis inhibitor, Z-VAD-FMK [17,18], and the ferroptosis inhibitors, Ferrostatin-1 [8,19] and Liproxstatin-1 [20], on Orf7b-induced cell death. Z-VAD-FMK, Ferrostatin-1, and Liproxstatin-1 partially alleviated apoptosis and ferroptosis, respectively (Figure 4F–H), in a concentration-dependent manner. These results suggest that Orf7b induces cell death by both apoptosis and ferroptosis.

### 2.4. c-Myc Regulates Orf7b-Induced Apoptosis and Ferroptosis

In line with the total RNA seq results, c-Myc has been previously implicated in inducing apoptosis and ferroptosis [21,22,23,24,25]. Hence, we determined whether Orf7b-induced cell death is regulated through c-Myc. First, Orf7b overexpression increased c-Myc protein expression (Figure 5A). We also looked at the protein expression of tumor necrosis factor alpha (TNF-α) and found it to be elevated in Orf7b-overexpressed cells (Figure 5B). To confirm the regulatory role of c-Myc in Orf7b-induced apoptosis and ferroptosis, we depleted c-Myc using the CRISPR-Cas9 knockout technique (Figure 5C, Appendix A). Overexpression of Orf7b in c-Myc^−/−^ cells did not change the protein levels of either cleaved caspase 8 or GPX4 as compared with the control (Figure 5D,E). In addition, knockout of c-Myc ablated Orf7b-induced active caspase 8, while preserving GPX4 levels in the Orf7b-overexpressing group. Knocking out c-Myc attenuated Orf7b-induced cell death, as confirmed by the LDH assay (Figure 5F), and reduced the intracellular ROS levels, assessed using the DCFDA assay (Figure 5G). These data suggest that Orf7b-induced apoptosis and ferroptosis are mediated through the induction of c-Myc.

### 2.5. Orf7b Induces Lung Injury via c-Myc Mediated Apoptosis and Ferroptosis in a Mouse Model

To confirm the in vivo relevance of our findings, we used intranasal administration of adenoviral-associated serotype 9 vectors to overexpress Orf7b in the presence or absence of c-Myc in mouse airways. Mice were divided into four groups (*n* = 8): wild type, AAV, AAV-Orf7b, and AAV-c-MYC shRNA then AAV-Orf7b [26,27]. Packaged viral vectors (5 × 10^11^ viral genome copies/mouse) were intranasally introduced into mouse lung for 7 days. For c-Myc knockdown, AAV-c-Myc shRNA constructs were instilled over 7 days. H&E staining of lung tissue sections showed that Orf7b overexpression induced severe lung injury with substantial infiltration of the inflammatory cells (Figure 6A, third panel). Depletion of c-Myc attenuated Orf7b-induced lung injury (Figure 6A, fourth panel). Increased cellularity of bronchoalveolar lavage fluid (BALF) in response to Orf7b overexpression was absent in c-Myc-depleted mice (Figure 6B). Similarly, depleting c-Myc prevented an Orfb7-induced increase in the BALF total protein content (Figure 6C). These data indicate that Orf7b is a potent viral factor that causes lung inflammation and injury, and that c-Myc is essential for the pathogenic effects of Orf7b.

We conducted fluorescent immunostaining to demonstrate whether Orf7b induced apoptosis and ferroptosis in vivo. Overexpression of Orf7b increased lung caspase 8 levels and decreased GPX4 expression (Figure 7A,B, lower middle panels), consistent with our findings in the BEAS-2B cell model. Depletion of c-Myc prevented both Orf7b-induced apoptosis and ferroptosis (Figure 7, lower panels). Expression of Orf7b was confirmed using qPCR analysis of whole lung RNA extracts (Appendix A). Knockdown of c-Myc was confirmed using immunoblotting analysis (Figure 7E first panel, Figure 7F). Overexpression of Orf7b increased the cleaved caspase 8 level (Figure 7E third panel, Figure 7G) and decreased the GPX4 level in the mouse lung tissues (Figure 7E fifth panel, Figure 7H), but these responses were absent with c-Myc depletion (Figure 7E, third and fifth panels, Figure 7F,G). These in vivo results confirm that Orf7b promoted lung inflammation, apoptosis, and ferroptosis, all of which are induced by c-Myc.

## 3. Discussion

Diffuse alveolar damage (DAD) is one of the important pathologic features reported in the autopsies of COVID-19 patients [28,29,30,31]. The SARS-CoV-2 factors that cause lung inflammation and damage with subsequent acute lung injury and acute respiratory distress syndrome are not fully understood. In this study, we globally profiled the cytotoxic effects of SARS-CoV-2 proteins in the context of lung epithelial cells. We found that Orf7b was one of the top ranked cytotoxic proteins. Orf7b induced statistically significant cell death in cultured lung epithelial cells, as well as pulmonary inflammation, injury, and cell death in vivo. The relatively small increase in the OD values and the percentage of cell death is possibly due to the approach of introducing viral factors into the cells that cause a high amount of cell death in the control groups. Mechanistically, the findings demonstrate that Orf7b requires the induction of c-Myc signaling and transactivation to promote injury and cell death, as depletion of c-Myc prevented Orf7b-promoted apoptosis. Remarkably, in the absence of c-Myc, Orf7b failed to initiate pulmonary edema and cellular infiltration, as well as epithelial injury, which are all hallmarks of acute lung injury caused by viral infection.

The predicted molecular weight of Orf7b is 5.1 kDa [32] and it contains 43 amino acid residues with enrichment in leucine (25.58%) and phenylalanine (13.95%) [32]. The sequence identity of SARS-CoV-2 Orf7b is 88% of SARS Orf7b, with a fully conserved transmembrane domain, two predicted helices, one helix–helix interaction domain, one beta turn, and one gamma turn without channel formation [32]. Orf7b, along with other viral factors, is reported to suppress Type-1 IFN responses [33,34]. Recombinant SARS-CoV-2 lacking the Orf7b protein (ΔOrf7b rSARS-CoV-2) caused a reduction in lung titer and marginally protected infected K18 hACE2 transgenic mice from death (25% reduction in comparison with the wild-type virus) [35]. Yet, the mechanisms by which Orf7b contributes to poor outcomes is unclear. Our data indicate a more pronounced role of Orf7b in causing lung injury through the promotion of both apoptotic and ferroptotic cell death. Activation of both cell death pathways required expression of the central regulator c-Myc, thus establishing a novel role of Orf7b (and c-Myc) in the pathogenesis of lung injury.

Of the 29 predicted proteins expressed by SARS-CoV-2, Orf3a, Orf6, Orf7a, Nsp6, spike, and M proteins have been reported to induce apoptosis in different cell lines [4,36,37,38,39]. Orf3a induces apoptosis in Vero-E6 and HEK-293T cells via caspase 3 activation, but this activation is weaker than that of SARS-CoV Orf3a [4,36]. A recent report has also implicated Orf3a in inducing ferroptosis via the Keap1-NrF2 axis [40]. Orf6, Orf7a, and NSP6 exhibit high levels of toxicity in human HEK-293T cells and localize at the membrane in COS-7 cells [4,37]. Spike protein has been shown to induce apoptosis in THP-1 macrophages by elevating TNF-α and MHC-II [38]. M protein triggers mitochondrial apoptosis by inducing B-cell lymphoma 2 (BCL-2) ovarian killer (BOK) [39]. In contrast to the Orf8 of SARS-CoV, which induces apoptosis, the Orf8 of SARS-CoV-2 has been reported to induce endoplasmic reticulum (ER) stress by activating Inositol-Requiring Enzyme 1 (IRE1) [4,41,42]. Although one study has reported that Orf7b is not toxic in HEK 293 T cells [37], it has been reported in another study that Orf7b induces TNF-α-mediated apoptosis in Vero-E6 and HEK 293T cells [13], which aligns with the function of Orf7b in SARS-CoV [14]. Interestingly, TNF-α-mediated apoptosis of CD4+ T cells has also recently been shown to contribute to SARS-Co-V2-induced lymphopenia [43]. A study has been published investigating the role of individually expressed Orf7b and has reported, using transcriptome and bioinformatic analysis, that Or7b disrupts several important cellular networks and deregulates several important host pathways, such as metabolism, cell adhesion, and immune response [34]. However, the cytotoxic aspect of Orf7b in the context of lung epithelial cells has not been investigated fully, which we have done in this study.

Overexpression of c-Myc has been reported to repress several survival pathways linked to extrinsic apoptosis [21]. In this study, we identified c-Myc as the transcriptional regulator of Orf7b-induced extrinsic or death-receptor-mediated apoptosis. In a previous study, it was found that there is a correlation between tumor necrosis factor alpha (TNF-alpha)-related apoptosis-inducing ligand (TRAIL) and c-Myc activation, with FLIP (Fas-associated death domain-like IL-1β-converting enzyme inhibitor protein) being the target of c-Myc-mediated transcriptional repression [44]. The reduced expression of FLIP, which is also a caspase 8 inhibitor, leads to caspase 8 activation, thereby inducing apoptosis [17,44]. Thus, previous studies point to the role of FLIP in c-Myc-mediated apoptosis. It has been reported that there is an increase in sensitivity to TNF-α-mediated cellular cytotoxicity on account of c-Myc overexpression [45]. Thus, our study reporting c-Myc upregulation in Orf7b-induced cytotoxicity links well with the study by Yang et al., showing the involvement of TNF-α in Orf7b-induced apoptosis [13]. Additionally, we also independently confirmed in this study that TNF-α was indeed upregulated in Orf7b overexpressed cells, thereby confirming the hypothesis.

It has also been hypothesized that ferroptosis may contribute to COVID-19 multiorgan damage failure [12]; however, data to support this are modest. Ferroptosis is a more recently identified type of cell death involving iron-dependent generation of hydroperoxy phospholipids through enzymatic or non-enzymatic means. GPX4, a peroxidase selective to the neutralization of lipid peroxides, is critical for the metabolism of these hydroperoxides to their neutral alcohols [8,9]. Ferroptosis also associates with increases in reactive oxygen species (ROS) which could also contribute to ferroptosis [8,9,10,11]. Ferrostatin-1, a ferroptosis inhibitor, decreases ferroptotic cell death through unclear mechanisms [8,19]. The involvement of ferroptosis in cell death due to SARS-CoV-2 infection has been reported in sinoatrial pacemaker cells, placenta, and endothelial cells [45,46,47]. Sinoatrial node pacemaker cells can be infected by SARS-CoV-2, where downregulation of GPX4 leads to ferroptotic cell death [46]. Placentas from 23 SARS-CoV-2-infected pregnant females were reported to express elevated levels of ACSL4 (acyl-CoA synthetase long-chain family member 4), an important component in the generation of ferroptotic phospholipids, thereby suggesting a role of ferroptosis infected human placenta [47]. A studying comparing the effects of serum from COVID-19 patients who survived versus those who did not on human endothelial cells found that the sera of the non-survivors caused significantly elevated lipid peroxidation and expression levels of important ferroptosis markers [48].

This is one of the first studies to identify SARS-CoV-2-induced ferroptosis in lung epithelial cells and in an in vivo lung model, while also identifying a transcription factor that may activate it. As with apoptosis, we show that Orf7b led to increased ferroptosis through the induction of c-Myc. c-Myc has been implicated in previous studies in apoptotic and ferroptotic pathways in different cellular contexts [21,22,23,24,25]. Some studies have reported the role of c-Myc in inhibiting ferroptosis [22,23], but the mechanisms for inducing ferroptosis in epithelial cells, as well as the role of c-Myc in GPX4 downregulation, are unclear. However, c-Myc is involved in the induction of genes responsible for coordinating iron metabolism, such as natural resistance-associated macrophage protein 1 (NRAMP1), transferrin receptor 1 (Tfr1), iron regulatory protein 2 (IRP2), and ferritin [21,49,50], which could contribute. It also remains to elucidate the mechanism of c-Myc activation by Orf7b. P53 is known to repress c-Myc by histone deacetylation mechanism [51]. The bioinformatic analysis of Orf7b-transduced cells in this study predicted the repression of p53 (Figure 3C,D). Therefore, repression of p53 on account of Orf7b transduction leading to the activation of c-Myc may be a potential mechanism, which can be the basis of future studies.

In conclusion, our data demonstrate novel parallel induction of c-Myc regulation of both apoptosis and ferroptosis following Orf7b exposure (Figure 8). Further elucidation of the mechanisms regulating these programmed cell death pathways in response to SARS-CoV-2 could identify novel targets for intervention in more severe COVID-19.

## 4. Materials and Methods

### 4.1. Cell Line and Reagents

Human lung epithelial cells (Beas-2B) were cultured in HITES medium (500 mL DMEM/F12, 2.5 mg transferrin, 10 μM hydrocortisone, 2.5 mg insulin, 10 μM Beta-estradiol, 2.5 mg sodium selenite, 10 mM HEPES, 2 mM L-glutamine) containing 10% fetal bovine serum (FBS). Adenocarcinomic human alveolar basal epithelial cells (A549) were cultured in F-12K medium (CAT#: 30-2004, ATCC), supplemented with 10% FBS. Primary human small airway epithelial cells (HSAEC) were cultured using bronchial epithelial growth kit (CAT#: PCS-300-040, ATCC). The cells were maintained at 37 °C, 5% CO_2_. The following were purchased from Invitrogen Technologies, Carlsbad, CA, USA: V5 antibody (Cat#: 46-1157), TOP10 competent cells (CAT#: C404006), the pcDNA3.1-V5-His-TOPO cloning kit (CAT#: K490001), and mouse anti-human IgG (H + L) secondary antibody (Cat#: 31420). Strep-tag II antibody was purchased from Abcam, Eugene, OR, USA (CAT#: ab76949). GPX4 mouse monoclonal antibody was purchased from Santa Cruz Bio, Dallas, TX, USA (CAT#: sc-166570). Caspase 8 rabbit antibody (CAT#: 56116SS) was purchased from Novus Biologicals, Centennial, CO, USA. Antibodies against Cleaved caspase 8 (CAT#: 9748), Caspase 9 (CAT#: 9502S), and β-actin (CAT#: A5441) were purchased from Cell Signaling Technology, Danvers, MA, USA. c-Myc mouse antibody was purchased from Biolegend, San Diego, CA, USA (CAT#: 626804) and Novus Bio, Centennial, CO, USA (CAT#: NB600-302SS). Goat anti-rabbit IgG-HRP conjugate (CAT#: 1706516) and goat anti-mouse IgG-horseradish peroxidase (HRP) conjugate (CAT#: 1706516) were purchased from Bio-Rad, CA, USA. LDH cytotoxicity detection kit was obtained from Takara Bio, San Jose, CA, USA (CAT#: MK401). RNA isolation and purification kit was purchased from Qiagen, Germantown, MD, USA (Cat. # 74004). Ferroptosis inhibitors Ferrostatin-1 (CAT#: A13247) and apoptosis inhibitor Z-VAD-FMK (CAT#: A12373) were purchased from AdooQ Bioscience, Irvine, CA USA. Liproxstatin-1 (Cat#: 17730) was purchased from Cayman Chemicals, Ann arbor, MI, USA. ROS Detection Cell-Based Assay Kit (DCFDA) was purchased from Cayman Chemical, Ann Arbor, MI, USA (Cat# 601520). Human Myc CRISPR/Cas9 KO Plasmids were purchased from Santa Cruz Biotechnologies, Inc., Dallas, TX, USA. (CAT# sc-400001-KO-2 and sc-400001-HDR-2). AAV9 control vector (VB220706-1168kuz), Orf7b expression (VB220705-1410adz), and ShRNA for mMyc (VB900131-5919nay) were obtained from VectorBuilder, Chicago, IL, USA. The SARS-CoV-2 plasmids, AAV9 packaging plasmid (CAT#: 112865), and helper plasmid (CAT#: 112867) were purchased from Addgene, Watertown, MA, USA. AAV purification kit was obtained from Takara Bio (CAT#: 6676). Goat anti-Mouse IgG (H + L) Cross-Adsorbed Secondary Antibody, Cyanine5 (CAT#: A10524), Goat anti-Rabbit IgG (H + L) Cross-Adsorbed Secondary Antibody, Cyanine5 (CAT#: A10523), and Trypan blue solution (0.4%) (CAT#: 15250061) were purchased from Thermo Fisher Scientific, Waltham, MA, USA.

### 4.2. Plasmid Transfection

Molecular cloning of viral genes and plasmid preparation were performed previously [52]. For assessing cytotoxicity, Beas-2B and A549 cells were transfected with the plasmids, using Lipofectamine-3000 (Cat # L3000001, Thermo Fisher Scientific, Waltham, MA, USA) as per the manufacturer’s protocol. The cells were cultured in a final volume of 2 mL HITES medium for 24 h.

### 4.3. RNA Sequencing and Analysis

Beas-2B cells overexpressed with empty plasmid were used as the control group and cells overexpressed with Orf7b plasmid were used as the treatment group (three samples per group). RNA isolation and purification were performed using the Qiagen RNeasy Micro Kit (Cat. # 74004, Germantown, MD, USA) as per the manufacturer’s protocol. Library preparation and RNA sequencing were performed at Medgenome Inc., CA, USA. The library was prepared using the Takara SMARTer Stranded Total RNA-Seq Kit v2—Pico Input Mammalian and the sequencing run performed using NovaSeq (PE100/150) (Illumina, San Diego, CA, USA), with 30 million total reads per sample (library preparation and sequencing run performed at Medgenome, Inc., San Diego, CA, USA.). The FASTQ data files were pre-processed using CLC Genomics Workbench 21 (Qiagen, Germantown, MD, USA). The differentially expressed genes were analyzed using ENRICHR, DAVID, and IPA. The threshold FDR *p*-value used was 0.05 and minimum fold change was 1.5.

### 4.4. LDH and Trypan Blue Assays

24 h after the viral plasmid transfection, cell supernatants from the control and treatment groups were collected for the LDH assay. The assay was performed according to the manufacturer’s protocol (Takara Bio, San Jose, CA, USA, CAT#: MK401). Absorbance of optical density (OD) was measured using an automated spectrophotometer. For the Trypan Blue assay, the cell suspension was mixed with the blue dye in equal amounts and the viable cell count was performed using an automated cell counter.

### 4.5. Intracellular ROS Assay

Cells transfected with control and Orf7b plasmids were cultured at 30,000 per well in 96-well plate overnight. The cell media were removed the next day and 100 µL of 1× phosphate-buffered saline (PBS) was added per well. After removing the buffer, the cells were stained with dichlorodihydrofluorescein diacetate (DCFDA) solution (5 μM diluted in PBS) and incubated in the dark at 37 °C for 40 min. DCFDA was removed from the wells and 100 µL/well of 1× PBS was added. H_2_O_2_ was used as positive control. Fluorescence measurement was done using fluorescence plate reader at 485/535 nm in endpoint mode.

### 4.6. Lipid Peroxidation Detection

Lipid peroxidation was assessed by measuring Malondialdehyde (MDA) content using an MDA assay kit (CAT# MA-MDA-2, Ray Biotech, Peachtree Corners, GA, USA) as per the manufacturer’s protocol. Briefly, 24 h after treatment, cells were washed twice with cold PBS and lysed and solubilized in a lysis buffer containing BHT (100× BHT solution added to 1× lysis buffer). The extracts transferred to microcentrifuge tubes were rocked gently at 4 °C for 30 min, followed by centrifugation at 14,000× *g* for 10 min. 100 µL of lysate samples and MDA standards were added to separate microcentrifuge tubes. 100 µL of sodium dodecyl sulfate (SDS) solution was added to each tube, mixed thoroughly and incubated at room temperature for 5 min. 250 µL of thiobarbituric acid (TBA) reagent was added to each sample and standard and incubated at 95 °C for 1 h. The tubes were then put on ice for 5 min, after which they were centrifuged at 1600× *g* for 10 min. 200 µL of supernatant from each sample and standard was added to a 96-well microplate and the absorbance was read immediately at 532 nm.

### 4.7. CRISPR/Cas9 Knockout

The transfection-ready knockout plasmids obtained from Santa Cruz Biotechnologies, Inc., Dallas TX, USA. (CAT# sc-400001-KO-2 and sc-400001-HDR-2) were used for the knockout studies. Solution A was prepared by diluting plasmid DNA in a plasmid transfection medium and solution B was prepared by diluting Ultracruz transfection reagent in a plasmid transfection medium. Solution A was added to solution B dropwise and the solution mix was vortexed immediately and incubated for 20–30 min. Beas-2B cells were cultured overnight in 6-well plates and 200,000–300,000 per well were transfected with Orf7b and control plasmids as per the treatment groups. The knockout solution mix was added to the c-Myc knockout/Orf7b treatment group and gently mixed by swirling the plate. 24–48 h after incubation, the successful knockout was confirmed using western blot analysis.

### 4.8. Western Blotting

Immunoblotting was performed as described previously [53,54]. Briefly, cells were collected and lysed with a lysis buffer (1× PBS, 0.5% Triton X-100, protease inhibitor). The cell lysates were sonicated and the debris was cleared by centrifugation at 13,000 rpm for 10 min at 4 °C. The protein concentrations of the samples were then determined using the Lowry assay. Protein separation was performed using sodium dodecyl sulfate-polyacrylamide gel electrophoresis (SDS-PAGE), followed by protein transfer onto a nitrocellulose membrane. The membranes were blocked for 45 min with 5% (*w*/*v*) non-fat dry milk solution (prepared using a Tris-buffered saline). The membranes were then incubated overnight at 4 °C, with the specific primary antibodies of appropriate dilutions. On the following day, the membranes were incubated with HRP-conjugated secondary antibodies at a 1:5000 to 1:10,000 dilution for 1 h at room temperature. The bands were visualized with enhanced chemiluminescence (ECL) and quantification was performed using the Ultrachemi imaging system (Bio-Rad, CA, USA).

### 4.9. Animal Models

The animal experiments were approved by the Institutional Animal Care and Use Committees at the University of Pittsburgh. The mice used for the experiment were of C57BL/6 breed, 6 to 8 weeks old. The total number of mice in the study, as well as number of mice per group, was calculated as per [26,27]. The virus was packaged as per the protocol described in [53]. AAV purification was performed as per the manufacturer’s protocol. The mice were administered an intranasal dose of 5 × 10^11^ vg of the respective AAVs and euthanized 7 days post operation under ABSL2.

### 4.10. Immunofluorescence and H&E Staining

Immunofluorescence staining was performed using lung paraffin sections from mice of WT, C, Orf7b, and c-MycKD + Orf7b groups to check the protein expression. 5-μm thick lung sections were embedded in paraffin and incubated overnight at 60 °C. They were then deparaffinized in xylene and the slides were hydrated in series in 100%, 95%, and 80% ethanol, followed by a wash in double-distilled water. The dewaxed slides were then treated with 50× DAKO high PH buffer (CAT#: K800421-2; Agilent Technologies, Santa Clara, CA, USA) in a pressure cooker for 20 min at 105 °C and cooled to room temperature. The slides were then incubated with 300 mM glycine for 30 min followed by overnight incubation at 4 °C with Mouse on Mouse (MOM) incubating reagent (CAT#: MKB-2213-1). The slides were washed with PBS for 2 min two times the following day and incubated with blocking solution (BSA 2%. Triton 0.1%, Tween 0.1%) for 30 min. The slides were then incubated with primary antibodies overnight at 4 °C, followed by incubation with goat anti-mouse IgG (H + L) Cyanine5 and goat anti-rabbit IgG (H + L) Cyanine5 conjugated secondary antibodies for 1 h. Finally, the slides were counterstained with 4′,6-diamidino-2-phenylindole (DAPI) and the images were captured using Keyence All-In-One Fluorescence Microscope (BZ-X810, Itasca, IL, USA). H&E stained slides obtained from the Division of Neuropathology, University of Pittsburgh, were imaged using Rebel Microscope (RBLT3000, Echo, San Diego, CA, USA).

### 4.11. Statistical Analyses

All data are presented as means ± standard deviations (SD). All statistical analyses were performed using GraphPad Prism 9.4.0. Continuous variables between the two groups were compared using an unpaired *t* test. A *p* value of <0.05 was considered statistically significant. A one-way ANOVA was performed for three or more variables.

## Figures and Tables

**Figure 1 ijms-25-01157-f001:**
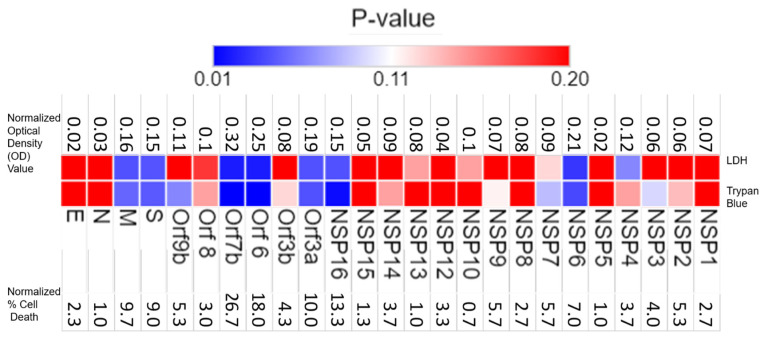
Global profiling SARS-CoV-2 proteins for cytotoxicity. The upper panel displays the results from the profiling using the LDH assay, while the lower panel displays the results from Trypan Blue assay. The heatmap corresponding to the *p*-values shows the range of 0.01 and 0.2, with extreme red representing *p*-values greater than 0.2 and extreme blue representing *p*-values less than 0.01. The values above the upper panel represent the absorbance values of the treatment groups adjusted by control groups. The values below the lower panel represent the difference in the percent cell death between the treatment and control groups. Results are representative of three independent experiments. *p*-values were calculated using a student *t*-test. The heatmap plotted was using Morpheus, https://software.broadinstitute.org/morpheus (accessed on 18 April 2023).The data are from three independent biological experiments.

**Figure 2 ijms-25-01157-f002:**
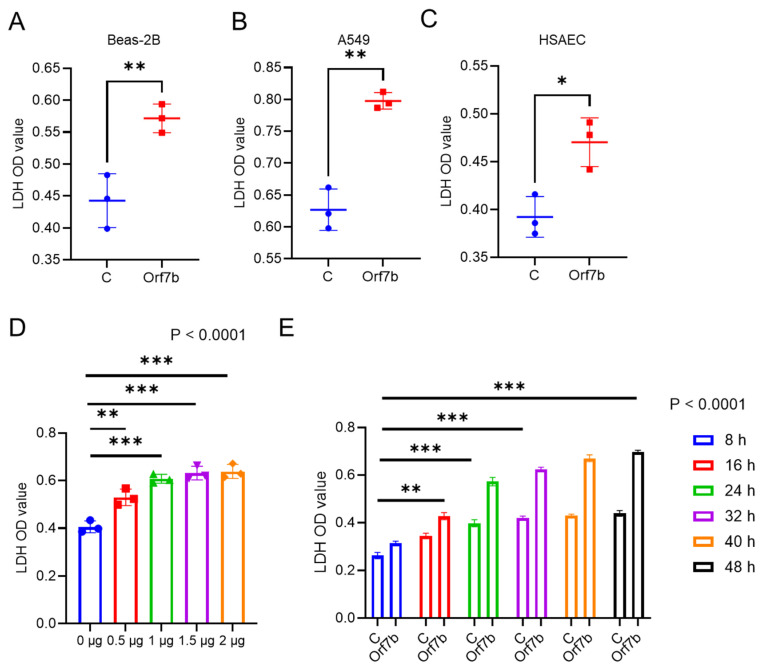
Orf7b induced cytotoxicity in lung epithelial cells. (**A**). Results of the LDH assay for Beas-2B cells. (**B**). Results of the LDH assay for A549 cells. (**C**). Results of the LDH assay for primary lung epithelial cells (HSAEC). (**D**). Concentration–response plot using the LDH assay for Beas-2B cells showed an increase in cytotoxicity with the increase in the amount of Orf7b transduced. (**E**). The cytotoxic effect of Orf7b in Beas-2B cells over multiple timepoints revealed a steady increase in cell death over time. C: control group, Orf7b: Orf7b overexpressed group. *: *p* ≤ 0.05, **: *p* ≤ 0.01, ***: *p* ≤ 0.001. *p*-values between two groups were calculated using a student *t*-test. *p*-values between three or more groups were calculated using a one-way ANOVA. Results are representative of three independent experiments.

**Figure 3 ijms-25-01157-f003:**
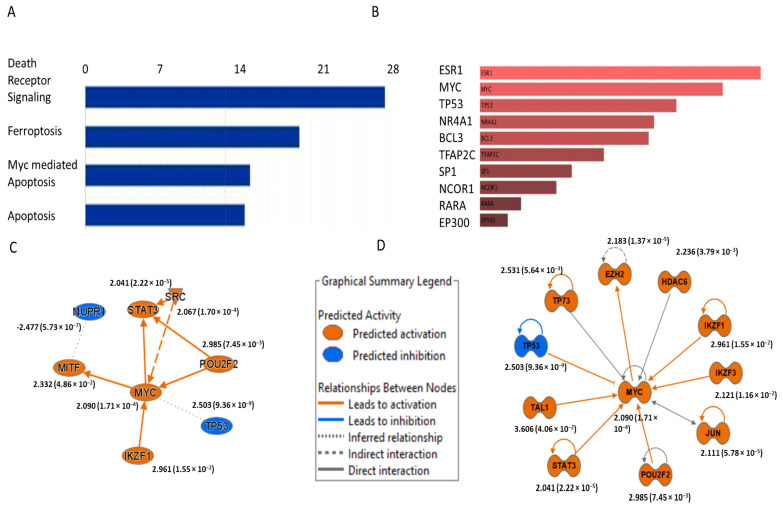
Functional enrichment analysis of the total RNA seq data revealed apoptosis and ferroptosis to be the implicated cell death pathways, with c-Myc as an important transcription regulator. (**A**) Analysis using IPA predicted the involvement of apoptosis and ferroptosis pathways. (**B**) Functional enrichment analysis using ENRICHR revealed c-Myc to be an important transcription regulator. (**C**,**D**) Graphical summary and upstream network analysis in IPA predicted the activation of the transcription regulator c-Myc, as well as its association with several important transcription factors. Values besides the gene symbols denote the predicted activation z-score and the values in parentheses indicate the respective *p*-values. The data are from three independent biological experiments.

**Figure 4 ijms-25-01157-f004:**
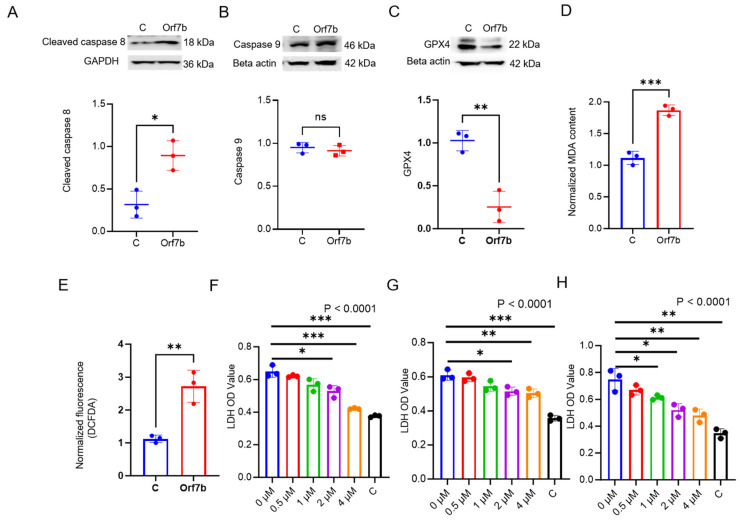
Orf7b expression activated cleaved caspase 8 but decreased GPX4. (**A**–**C**) Immunoblotting revealed upregulation of cleaved caspase 8 (**A**), no change in expression level for caspase 9 (**B**) and downregulation of GPX4 (**C**), with corresponding densitometric plots. (**D**) The MDA assay revealed that Orf7b induced statistically significant lipid peroxidationin Beas-2B cells compared with the control group. (**E**) The DCFDA assay revealed that there was significant increase in intracellular ROS levels in the Orf7b group compared with the control group. (**F**,**G**) Concentration response-based apoptosis inhibitor (Z-VAD-FMK) and ferroptosis inhibitor (Ferrostatin) studies using LDH assays showed partial alleviation of cell death in Orf7b-overexpressed Beas-2B cells, respectively. (**H**) A ferroptosis inhibitor, Liproxstatin-1, based inhibitor study confirmed the involvement of the ferroptosis pathway. For (**A**–**C**)—C: control group, Orf7b: Orf7b overexpressed group.*: *p* ≤ 0.05, **: *p* ≤ 0.01, ***: *p* ≤ 0.001, ns: not significant. *p* values between two groups were calculated using a student *t*-test. *p* values between three or more groups were calculated using a one-way ANOVA. The data are from three independent biological experiments.

**Figure 5 ijms-25-01157-f005:**
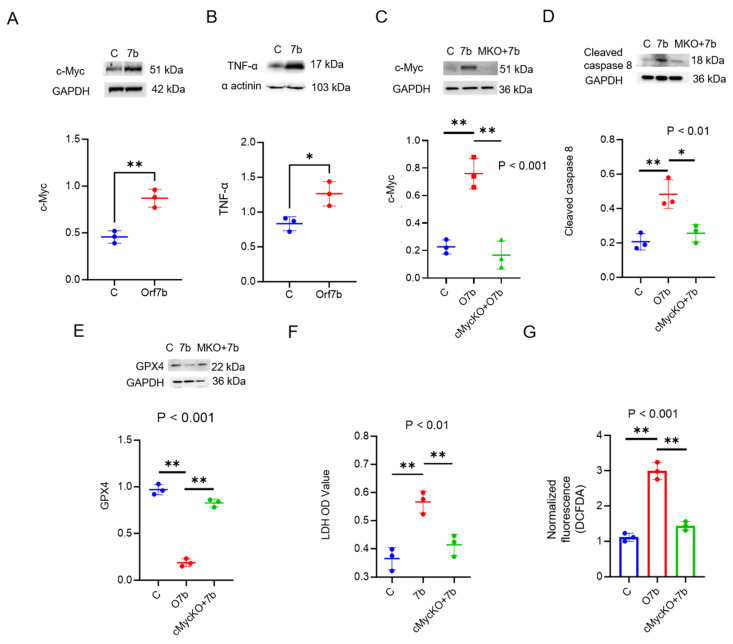
Orf7B upregulates c-Myc to modulate the activation of cleaved caspase 8 and the reduction of GPX4. (**A**–**D**) Immunoblotting revealed upregulation of c-Myc in Orf7b-transduced Beas-2B cells (**A**), upregulation of TNF-α in Orf7b-transduced Beas-2B cells (**B**). CRISPR/CAS9 knockout of c-Myc in Orf7b-transduced Beas-2B cells (**C**) implicated c-Myc as the critical regulator in apoptosis and ferroptosis by downregulating the expression of cleaved caspase 8 (**D**) and alleviating the expression of GPX4 (**E**). Corresponding densitometric plots for the immunoblots confirm the results (**A**–**E**). (**F**) The LDH assay confirmed the attenuation of cell death post c-Myc knockout. (**G**) The DCFDA assay revealed a decrease in intracellular ROS levels after knocking out c-Myc. For (**A**,**B**,**D**)—C: Control, 7b: Orf7b overexpression, M: c-Myc knockout + Orf7b overexpression. For the densitometric plots—C: control group, Orf7b: Orf7b overexpressed group *: *p* ≤ 0.05, **: *p* ≤ 0.01. *p*-values between the two groups were calculated using a student *t*-test. *p*-values between three or more groups were calculated using a one-way ANOVA. The results are representative of three independent experiments.

**Figure 6 ijms-25-01157-f006:**
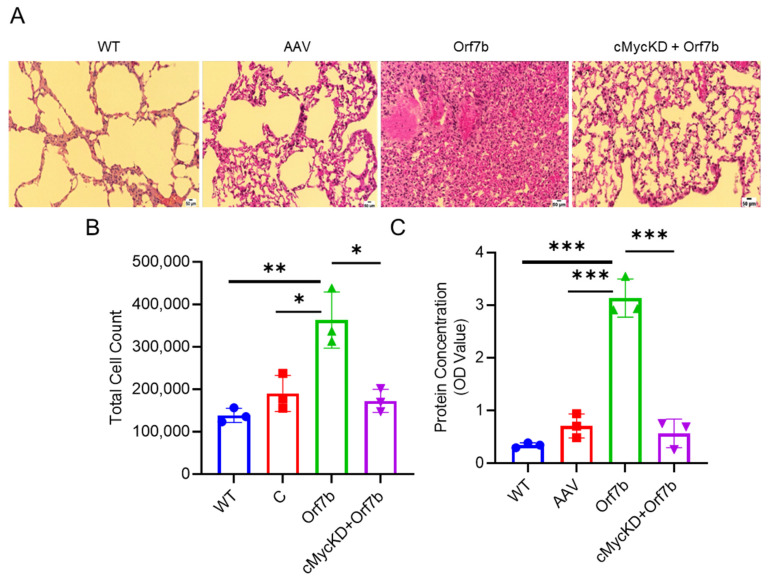
Overexpression of Orf7b induced lung injury in a mouse model. (**A**) H&E staining and subsequent light microscope imaging revealed differences in the lung morphologies of the mice belonging to WT, C, Orf7b, and c-MycKD + Orf7b groups (*n* = 8 per group). (**B**,**C**) Total cell count and BALF protein concentration were elevated for Orf7b group compared wih the other groups. WT: wild type group, AAV: control group, Orf7b: Orf7b overexpressed group, c-MycKD + O7b: c-Myc knockdown group overexpressed with Orf7b. *: *p* ≤ 0.05, **: *p* ≤ 0.01, ***: *p* ≤ 0.001. *p*-values between two groups were calculated using a student *t*-test. *p*-values between three or more groups were calculated using a one-way ANOVA. Scale bar: 50 µm.

**Figure 7 ijms-25-01157-f007:**
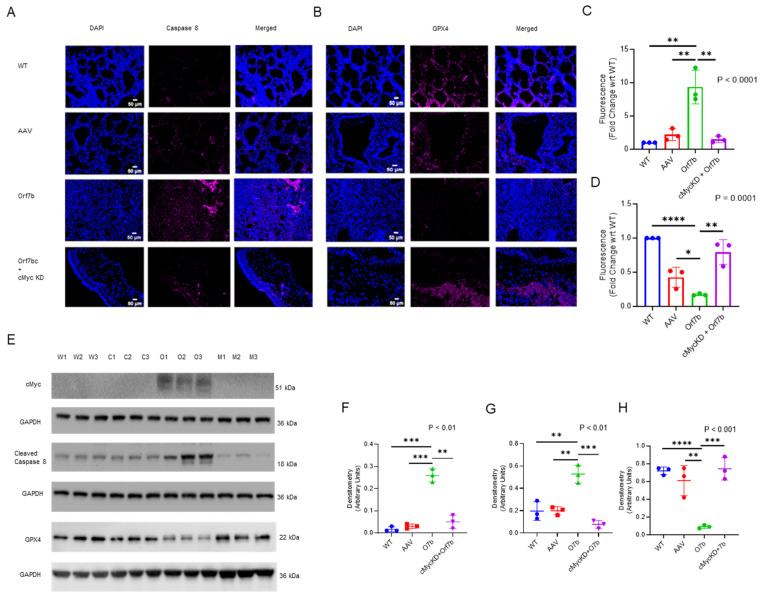
Orf7b-induced lung injury induced c-Myc and increased apoptotic and ferroptotic cell death. (**A**,**B**) Immunofluorescence staining showed caspase 8 activation in the Orf7b overexpressed group compared with the wild type, control, and c-Myc knockdown groups (*n* = 8 per group) (**A**) and GPX4 inhibition in the Orf7b-overexpressed group compared with the other three groups (**B**). Corresponding fluorescence intensity plots using ImageJ 1.54d are shown in (**C**,**D**), respectively. The in vivo mouse model confirmed the results from the in vitro study: Immunoblot results for the protein expression of c-Myc (**E**, first panel), cleaved caspase 8 (**E**, third panel), and GPX4 (**E**, fifth panel). The corresponding densitometric plots are also shown (**F**–**H**). For (**E**)—W: Wild type, C: Control, O: Orf7b overexpression, M: c-Myc knockdown + Orf7b overexpression. For (**A**–**D**,**F**–**H**)—WT: wild type group, AAV: control group, Orf7b-Orf7b-overexpressed group, c-MycKD + O7b: c-Myc knockdown group overexpressed with Orf7b. *: *p* ≤ 0.05, **: *p* ≤ 0.01, ***: *p* ≤ 0.00, ****: *p* ≤ 0.001. *p*-values between the two groups were calculated using a student *t*-test. *p*-values between three or more groups were calculated using a one-way ANOVA. Scale bar: 50 µm. It would be good to have the actual conditions on (**C**).

**Figure 8 ijms-25-01157-f008:**
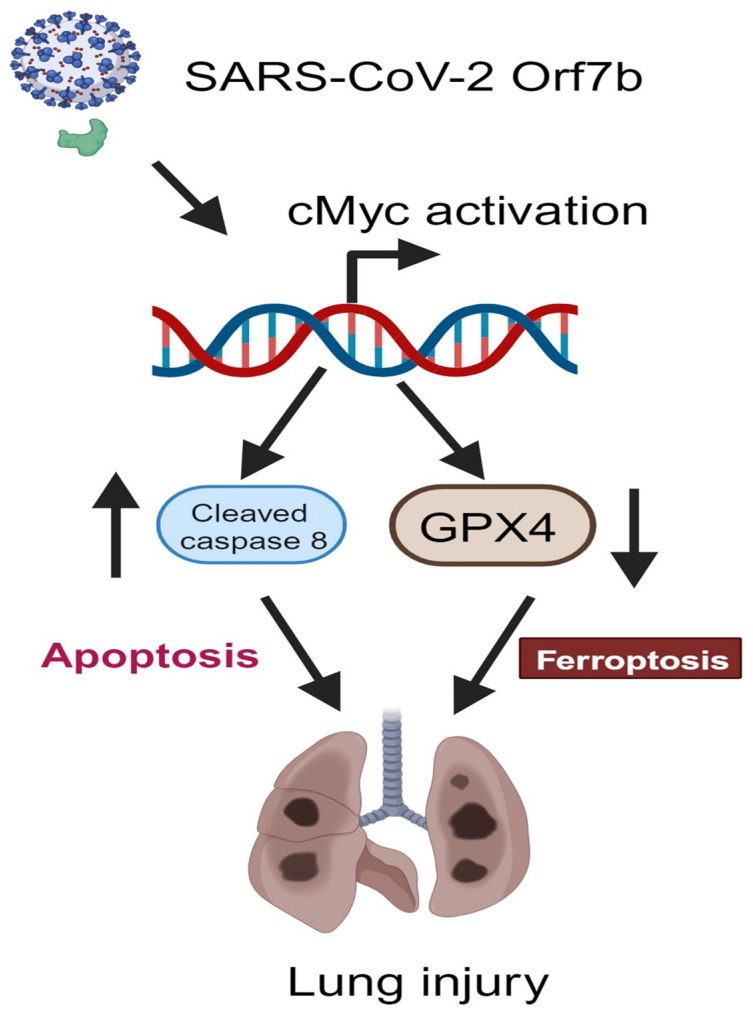
Scheme for the Orf7b-induced cell death pathways, apoptosis, and ferroptosis, in lung epithelial cell and mouse models. The SARS-CoV-2 accessory protein Orf7b induces cell death pathways in both extrinsic apoptosis and ferroptosis. Orf7b increases c-Myc at the protein level to modulate apoptosis and ferroptosis. Increased apoptosis and ferroptosis thus promote acute lung injury (Figure created using Biorender.com).

## Data Availability

The differentially expressed gene data set has been uploaded with the Appendix A. Any other data are available upon reasonable request.

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
