# Peer review of "SARS-CoV-2 Accessory Protein Orf7b Induces Lung Injury via c-Myc Mediated Apoptosis and Ferroptosis"

_ijms, 2024, doi:10.3390/ijms25021157_

Round 1

Reviewer 1 Report (Previous Reviewer 2)

Comments and Suggestions for Authors

The authors have addressed my concerns, I have no more comments.

Author Response

We thank the reviewer for the suggestive commentary that make the quality of the manuscript better.

Reviewer 2 Report (Previous Reviewer 1)

Comments and Suggestions for Authors

The authors have addressed all my points. The manuscript is now convincing. Only minor revisions are required.

1.     In the introduction the authors state “…both unprogrammed (necrotic) and programmed cell death (NEED REFS) [4]. Programmed cell death can be further classified as apoptotic and  non-apoptotic [7]. Non-apoptotic programmed cell death type consists of ferroptosis, anoikis, netosis, pyroptosis and necroptosis (and perhaps others) [7].“ There is a difference between programmed and regulated cell death. See reference [7]: “Programmed cell death (PCD). Particular form of RCD that occurs in strictly physiological scenarios, i.e., it does not relate to perturbations of homeostasis and hence does not occur in the context of failing adaptation to stress.” Please, change the sentence to “…both accidental (necrotic) and regulated cell death (NEED REFS) [4]. Regulated cell death can be further classified as apoptotic and non-apoptotic [7]. Non-apoptotic regulated cell death type consists of ferroptosis, anoikis, netosis, pyroptosis and necroptosis (and perhaps others) [7].”

2.     Please, include in all figure legends the number of replicates (in case of representative experiments) or the number of experiments performed.

3.     Fig.2E and Fig. 4F-H: which changes are statistically significant? Please, include bars as done in Fig. 2D

4.     Page 8, line 257: typo “similarly”

5.     Page 8, line 275: Please, correct to “Fig. 7 lower panels”

6.     Page 10, line 366: typo “apoptosis”

7.     Please, check the spaces between your words, examples: p10 line 340, line 342, line 346, p11 line 386, line 387…..

Author Response

The authors have addressed all my points. The manuscript is now convincing. Only minor revisions are required.

Response: We are grateful for the reviewer’s efforts that improved the quality of the manuscript.

  1. In the introduction the authors state “…both unprogrammed (necrotic) and programmed cell death (NEED REFS) [4]. Programmed cell death can be further classified as apoptotic and  non-apoptotic [7]. Non-apoptotic programmed cell death type consists of ferroptosis, anoikis, netosis, pyroptosis and necroptosis (and perhaps others) [7].“ There is a difference between programmed and regulated cell death. See reference [7]: “Programmed cell death (PCD). Particular form of RCD that occurs in strictly physiological scenarios, i.e., it does not relate to perturbations of homeostasis and hence does not occur in the context of failing adaptation to stress.” Please, change the sentence to “…both accidental(necrotic) and regulatedcell death (NEED REFS) [4]. Regulated cell death can be further classified as apoptotic and non-apoptotic [7]. Non-apoptotic regulated cell death type consists of ferroptosis, anoikis, netosis, pyroptosis and necroptosis (and perhaps others) [7].”

Response: We have followed the reviewer’s recommendation and made the changes as such.

  1. Please, include in all figure legends the number of replicates (in case of representative experiments) or the number of experiments performed.

Response:  We have listed the number of replicates of each experiment in the figure legends accordingly.

  1. Fig.2E and Fig. 4F-H: which changes are statistically significant? Please, include bars as done in Fig. 2D

Response:  We have showed the statistics among groups with bars.

  1. Page 8, line 257: typo “similarly”

Response:  We have changed the typo accordingly. 

  1. Page 8, line 275: Please, correct to “Fig. 7 lower panels”

Response: We have made the changes.

  1. Page 10, line 366: typo “apoptosis”

Response: We have changed the typo with corrected one.

  1. Please, check the spaces between your words, examples: p10 line 340, line 342, line 346, p11 line 386, line 387…..

 Response: We have rechecked the manuscript for the correct format throughout the text.

This manuscript is a resubmission of an earlier submission. The following is a list of the peer review reports and author responses from that submission.

Round 1

Reviewer 1 Report

Comments and Suggestions for Authors

In the manuscript ”SARS-CoV-2 accessory protein Orf7b induces lung injury via c-Myc-mediated apoptosis and ferroptosis” authored by Deshpande et al., the authors investigate the molecular mechanism(s) of Orf7b-mediated cell death in lung cells in vitro and in vivo. For in vitro studies, Orf7b was expressed in BEAS-2B cells. Expression of Orf7b resulted in a moderate, but statistically significant, cell death. Total RNA sequencing and subsequent bioinformatic analysis gave hint at possible cell death pathways, i.e. apoptosis and ferroptosis with c-Myc as a central and common and central regulator. The authors confirmed upregulation of c-Myc by Western blot analysis. Apoptosis was confirmed by cleavage of caspase 8, and ferroptosis by demonstrating downregulation of GPX4 protein as well as by pharmacological inhibition using a pan-caspase inhibitor and ferrostatin, respectively. The authors further revealed that c-Myc-k.o. prevented cell death, intracellular ROS-formation, cleavage of caspase 8 and downregulation of GPX4. Finally, the authors performed in vivo experiments in mice by intranasal administration of an adenoviral vector to express Orf7b in the absence or presence of an adenoviral shRNA construct allowing c-Myc knock-down. The authors present a model in which SARS-CoV-2 Orf7b induces c-Myc expression thereby inducing extrinsic apoptosis (cleavage of caspase 8) and ferroptosis by downregulation of GPX4 finally leading to lung injury. 

The authors address an important scientific issue, and their findings are relevant. The data are clearly presented and convincing. However, several points remain unclear and/or have to be improved. 

Major points

1.     Fig. 1: Orf7b induces a statistically significant, but nevertheless small increase in cell death. The authors should address this point in the discussion.

2.     Fig. 2 + 4: Cell death and protein levels are only investigated at one time point, i.e. 24 h after transfection. I suggest to extend the data and analyze the effects at different time points since this would maybe give a clue about the underlying mechanism.

3.     Fig. 4: The results with the inhibitors are not clear to me (E, F). Ferrostatin-1 nearly fully inhibits Orf7b-mediated toxicity. The sum of inhibition by Ferrostatin-1 and by Z-VAD-FMK is numerically more than 100 %. The authors have to discuss this. 

Furthermore, additional experiments are required to confirm ferroptosis. Measurement of lipid peroxidation is absolutely necessary as well as the use of additional inhibitors, such as liproxstatin-1 and an iron chelator, e.g. deferoxamine. 

4.     Fig. 7: I assume that the immunohistochemistry is not correctly labelled. Please, check “Cy5”. I guess, this is DAPI-staining?

5.     It would be easy to study FLIP and TNFa protein expression which would clarify the role of Orf7b in apoptosis in lung cells. 

6.     The discussion has to be extended. 

1. The authors should address the point that Orf7b is not toxic in HEK293T cells (Lee et al., 2020, Ref. 35 in the manuscript). 

2. Although the authors cite the papers, they should clearly mention, that c-Myc may also inhibit ferroptosis. 

3. The authors should discuss – if possible - potential mechanisms of (i) c-Myc-induction by Orf7b and (ii) downregulation of GPX4 by c-Myc. 

4. The authors should discuss differences and overlaps between their findings and the work by Garcia-Garcia et al., 2022 (Ref. 33 in the manuscript). 

Minor points

1.     Page 8, line 182, typo “intoduced”, please correct

2.     lines 186-188: I would prefer “in response to” instead of “caused by”  since “caused by” could also mean that the BALF was caused by Orf7b.

Author Response

Reviewer 1’s comment: 

The authors address an important scientific issue, and their findings are relevant. The data are clearly presented and convincing. However, several points remain unclear and/or have to be improved. 

Response: Thank you for summarizing and highlighting the importance of our work. We will try addressing all your concerns point by point as follows. 

Major points

  1. Fig. 1: Orf7b induces a statistically significant, but nevertheless small increase in cell death. The authors should address this point in the discussion.

Response: We have now included this point in the discussion in line 321. One of the reasons for this relatively smaller increase in cell death maybe due to the fact that the control vector itself induces substantial cell death. We have included this point in discussion.

  1. Fig. 2 + 4: Cell death and protein levels are only investigated at one time point, i.e. 24 h after transfection. I suggest to extend the data and analyze the effects at different time points since this would maybe give a clue about the underlying mechanism.

Response: We have now added the data analyzing the effect of Orf7b at multiple time points (Fig. 2 E). We obtained similar results.

  1. Fig. 4: The results with the inhibitors are not clear to me (E, F). Ferrostatin-1 nearly fully inhibits Orf7b-mediated toxicity. The sum of inhibition by Ferrostatin-1 and by Z-VAD-FMK is numerically more than 100 %. The authors have to discuss this. 

Response: We redid the inhibitor studies, reanalyzed the data and rectified the error. The inhibition by Z-VAD-FMK is more than Ferrostatin-1, and the sum of the inhibition is now less (Fig. 4E, F)

Furthermore, additional experiments are required to confirm ferroptosis. Measurement of lipid peroxidation is absolutely necessary as well as the use of additional inhibitors, such as liproxstatin-1 and an iron chelator, e.g. deferoxamine.

Response: We have now added the data for lipid peroxidation measurement using MDA assay as well as inhibition study using liproxstatin-1 (Fig. 4D, 4H).

  1. Fig. 7: I assume that the immunohistochemistry is not correctly labelled. Please, check “Cy5”. I guess, this is DAPI-staining?

Response: We have corrected the error in labelling in Fig. 7.

  1. It would be easy to study FLIP and TNFa protein expression which would clarify the role of Orf7b in apoptosis in lung cells. 

Response: We have included the western blot data for TNF-alpha (Fig. 5F). As expected TNF-α protein expression was upregulated in Or7b transduced cells compared to the control group. We have also expanded the discussion on FLIP. 

  1. The discussion has to be extended. 
  2. The authors should address the point that Orf7b is not toxic in HEK293T cells (Lee et al., 2020, Ref. 35 in the manuscript). 

Response: We have now addressed the point, in lines 353-356. Although it has been reported in reference number 35, that Orf7b is not toxic in HEK-293T cells, there has been another study published which reports the cytotoxicity of Orf7b in both HEK-293T and Vero-E6 cells (Reference number 13 in the manuscript), which also aligns with our results.

  1. Although the authors cite the papers, they should clearly mention, that c-Myc may also inhibit ferroptosis. 

Response: We have now stated this point in lines 398-399.

  1. The authors should discuss – if possible - potential mechanisms of (i) c-Myc-induction by Orf7b and (ii) downregulation of GPX4 by c-Myc.

Response: We have now included these points in the discussion, lines 398-408. 

  1. The authors should discuss differences and overlaps between their findings and the work by Garcia-Garcia et al., 2022 (Ref. 33 in the manuscript). 

Response: We have included this in the discussion, lines 357-362.

Minor points

  1. Page 8, line 182, typo “intoduced”, please correct

Response: We have corrected the typo.

  1. lines 186-188: I would prefer “in response to” instead of “caused by” since “caused by” could also mean that the BALF was caused by Orf7b.

Response: We have incorporated your suggestion.

Reviewer 2 Report

Comments and Suggestions for Authors

Deshpande et al. have investigated the role of Orf7b, an accessory of SARS-CoV2 virus. The authors have overexpressed several viral genes in a lung airway epithelial cell line and identified Orf7b as the top-ranked cytotoxic protein. Apoptosis and ferroptosis, both cell death pathways, were associated with cytotoxicity induced by overexpression of Orf7b and were transcriptionally mediated through cMyc. The authors have substantiated their invitro data in a mouse model using AAV overexpression of Orf7b and AAV-cMyc shRNA and showed that, indeed, overexpression of Orf7b has a profound effect on lung injury and inflammation and knockdown of cMyc rescue the phenotype. Overall, the study is very interesting, and the data is solid. I have the following comments.

1.     Fig 1 is very confusing to read. The authors should label the upper and lower panels within the figure. Also, in the figure, I assume the number indicates the p-value. That should also be labeled in the figure.

2.     The most confusing part is the involvement of two different cell death pathways. It is very highly unlikely that both pathway is equally involved. The authors have provided very good data with GPX4 as well as caspase8/3, but their inhibitor data is very weak. I don’t understand why the authors have used a suboptimal dose of both ferrostatin-1 (which is 10uM) and Z-VAD-FMK (which is 10uM). The dose used by the authors is too low to show the difference. This is very important data that may indicate the importance of one pathway over another.

3.     In the in vivo experiment was done for 7 days. The authors have ever performed a long-term study to examine if the overexpression of Orf7b affects the animals' mortality.

4.     Typos in the lines 54 and 284

Author Response

Reviewer 2’s comment:

Deshpande et al. have investigated the role of Orf7b, an accessory of SARS-CoV2 virus. The authors have overexpressed several viral genes in a lung airway epithelial cell line and identified Orf7b as the top-ranked cytotoxic protein. Apoptosis and ferroptosis, both cell death pathways, were associated with cytotoxicity induced by overexpression of Orf7b and were transcriptionally mediated through cMyc. The authors have substantiated their invitro data in a mouse model using AAV overexpression of Orf7b and AAV-cMyc shRNA and showed that, indeed, overexpression of Orf7b has a profound effect on lung injury and inflammation and knockdown of cMyc rescue the phenotype. Overall, the study is very interesting, and the data is solid. I have the following comments.

Response: We thank the reviewer for summarizing and highlighting the importance of our study. We will try our best to address your concerns point by point as below.

  1. Fig 1 is very confusing to read. The authors should label the upper and lower panels within the figure. Also, in the figure, I assume the number indicates the p-value. That should also be labeled in the figure.

Response: We have incorporated your suggestion and labeled the panels in Fig. 1 appropriately. The upper panel represents OD value adjusted to the control group, assessed using LDH assay, while the lower panel represents % cell death adjusted to the control group, assessed using trypan blue based cell counting method.

  1. The most confusing part is the involvement of two different cell death pathways. It is very highly unlikely that both pathways are equally involved. The authors have provided very good data with GPX4 as well as caspase8/3, but their inhibitor data is very weak. I don’t understand why the authors have used a suboptimal dose of both ferrostatin-1 (which is 10uM) and Z-VAD-FMK (which is 10uM). The dose used by the authors is too low to show the difference. This is very important data that may indicate the importance of one pathway over another.

Response: We redid the inhibitor studies using different range of concentrations and found the contribution of apoptosis pathway is more than ferroptosis pathway. We also confirmed the involvement of ferroptosis using an additional inhibitor Liproxstatin-1 as well as measured lipid peroxidation using MDA assay. We have included these data in Fig.4.

  1. In the in vivo experiment was done for 7 days. The authors have ever performed a long-term study to examine if the overexpression of Orf7b affects the animals' mortality.

Response: We indeed performed a 14-day study involving Orf7b overexpression in mice and found that there were no animal mortalities. The tissue morphology was also similar to the 7-day study.

  1. Typos in the lines 54 and 284

Response: We have now made the corrections.